# Learning Transparent Reward Models via Unsupervised Feature Selection

**Daulet Baimukashev**[1,*]**, Gokhan Alcan**[2]**, Kevin Sebastian Luck**[3]**, Ville Kyrki**[1]
[1]Aalto University   [2]Tampere University   [3]Vrije Universiteit Amsterdam

**Abstract:** In complex real-world tasks such as robotic manipulation and autonomous driving, collecting expert demonstrations is often more straightforward than specifying precise learning objectives and task descriptions. Learning from expert data can be achieved through behavioral cloning or by learning a reward function, i.e., inverse reinforcement learning. The latter allows for training with additional data outside the training distribution, guided by the inferred reward function. We propose a novel approach to construct compact and transparent reward models from automatically selected state features. These inferred rewards have an explicit form and enable the learning of policies that closely match expert behavior by training standard reinforcement learning algorithms from scratch. We validate our method's performance in various robotic environments with continuous and high-dimensional state spaces. Webpage: `https://sites.google.com/view/transparent-reward`.

**Keywords:** Inverse reinforcement learning, Reinforcement learning, Imitation learning, Reward learning, Robot learning

## 1   Introduction

Imitation learning (IL) involves learning policies from expert demonstrations. It has been successfully applied in many real-world robotics applications such as household tasks [1, 2], manipulation [3, 4, 5], and autonomous driving [6, 7]. It is more straightforward for humans to demonstrate task execution than to precisely specify task descriptions and/or learning objectives [8]. More broadly, IL can be addressed by either behavioral cloning [9] or inverse reinforcement learning (IRL) [10]. The first approach learns the mapping from state to action using supervised learning but suffers from compounding errors outside of the training distribution [11]. Reinforcement learning (RL) [12] benefits from exploration of the environment and learns through experience. RL has been applied in robotic applications [13, 14, 15, 16, 17], but it requires a well-defined reward function to solve the task. Manually specifying a reward function poses additional challenges for complex tasks, creating a need for the automatic construction of rewards from data. We propose to construct reward by automatically selecting its component features and retrieving reward using inverse reinforcement learning (IRL) setting [18, 10]. IRL benefits from expert demonstrations while utilizing more exploration.

One of the prominent approaches for learning rewards is maximum entropy inverse reinforcement learning (Max-ent IRL), which addresses the reward ambiguity problem using a probabilistic model of the behavior [19, 20, 21]. Max-ent IRL can learn an explicit reward model, but typically the features that compose the reward are specified beforehand. Both linear [22, 23, 24] and non-linear [25, 26] functions of features have been used to learn the reward. As depicted in Fig. 1 our method does not rely on hand-picked features as in current IRL methods, but finds relevant set of features from large candidate set. We learn *transparent* reward model such that the relationship between

---

[*]Corresponding Author

8th Conference on Robot Learning (CoRL 2024), Munich, Germany.

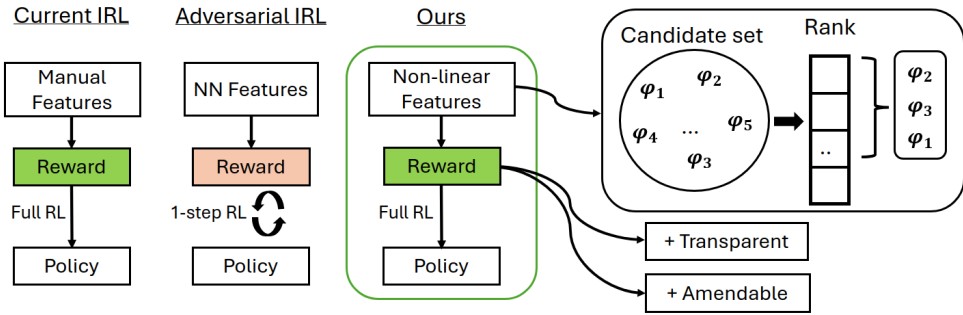

Figure 1: Current IRL methods require hand-picked features to use as reward components. Adversarial methods learn the reward using neural networks which are less transparent and not amendable. But, the proposed method finds automatically relevant set of state features and constructs transparent reward models.

state features and the resulting reward is direct, and reward function is compact and *amendable* if preferred.

In this work, we propose to construct a reward function from state features using unsupervised feature selection without requiring access to ground truth rewards. To achieve this, we use the maximum entropy formulation for stochastic policies, in which the probability of a trajectory is proportional to its cumulative reward. This approach mitigates the need for an exhaustive search for suitable features and produces an explicitly represented reward model. As only state features are used for reward components, this reward formulation is also referred to as *disentangled* rewards and is robust to changes in the environment dynamics, as presented in [27]. Our contributions are as follows:

1. We develop a method for learning a compact and transparent reward function for use in reinforcement learning algorithms.

2. We demonstrate the effectiveness of our proposed method through validation on various continuous control systems with high-dimensional, continuous state spaces.

## 2 Related Works

### 2.1 Feature construction for IRL

Inverse reinforcement learning methods can retrieve rewards if their structure is specified accordingly. Many works have considered manually designed features [28, 23, 20, 29, 30] for reward. While human-specified rewards are effective in some scenarios, given only demonstrations, it is challenging to design a reward function that matches the true intent or reward of an expert. The literature on automatically learning compact reward structures is limited. Reward construction from atomic features using regression trees was proposed in [31]. This method iteratively constructs the features and corresponding rewards. Another work [26] considered Gaussian processes for modeling nonlinear function approximation for the reward. A Bayesian approach using similar atomic features that constructs logical conjunctions of features was presented in [32]. Authors in [33] pre-train feature vector using self-supervised loss, but uses trajectory rankings to learn the reward function. Compared to previous approaches, we do not assume to have relevant atomic features or preference labels. Instead, we use states only from expert trajectories and consider nonlinear basis functions of states as reward components. This results in an explicitly represented reward model that consists of a small number of features.

## 2.2 Adversarial IRL

An alternative approach for imitation learning, effective in high-dimensional tasks, is enhanced with adversarial methods as presented in [27, 34, 35, 36]. These methods adopt the generative adversarial networks (GANs) [37] training scheme to generate trajectories similar to those of experts. The cost of the discriminator is usually referred to as a learned cost or reward. While adversarial methods achieve state-of-the-art performance for a wide range of applications [38, 39, 40], the retrieved reward function is a neural network and not an explicitly represented reward model.

## 3 Problem Statement

Reinforcement learning (RL) is an approach to solve Markov Decision Processes (MDP) defined as a tuple $\mathcal{M} = \langle \mathcal{S}, \mathcal{A}, \mathcal{R}, \mathcal{T}, \gamma \rangle$, where $\mathcal{S}$ is the set of states, $\mathcal{A}$ the set of actions, $\mathcal{R}$ is the reward function, $\mathcal{T}$ represents transition dynamics, and $\gamma$ is a discount factor. The solution represented by an optimal policy is defined as the one maximizing the expected discounted cumulative reward of the trajectory:

$$\pi^* = \arg\max_{\pi} \mathbb{E}_{p(s_{t+1}|s_t,a_t),\pi} \left[ \sum_{t=0}^{\infty} \gamma^t R(s_t) \right] \tag{1}$$

Here, $\gamma$ is a discount factor, transition probability $p(s'|s,a)$ is fixed but unknown, and the policy is learned to optimize this objective from experience.

Now we consider MDP setting, where in addition to transition probabilities, the task reward is unknown. Our objective is to reconstruct this reward function from observational data. As proposed in [10], we represent the reward as a linear combination of features which can consists of any nonlinear function of the state.

$$R(\mathbf{s}) = \theta^T \phi(\mathbf{s}) \tag{2}$$

where $\mathbf{s}$ is the state, $\phi \in \mathbb{R}^d$ is the feature function, mapping the state $\mathbf{s}$ to a $d$-dimensional feature vector, and $\theta \in \mathbb{R}^d$ is the weight vector of features. The reward construction problem then translates to finding suitable features and their corresponding weights.

## 4 Proposed Method

In this section, we present our method for learning a reward model, which consists of two aspects: finding the feature set and its weight vector. As ground truth rewards are not available, we use unsupervised feature selection by generating pseudo labels from expert data. Then, we adopt the max-ent IRL framework for learning feature weights. The overview of the algorithm is provided in Appendix C.

### 4.1 Trajectory probability under maximum entropy

According to the maximum entropy formulation for stochastic policies [19], the probability of a trajectory is proportional to the exponential of the reward of that trajectory. That is,

$$P(\tau_i|\theta) = \frac{e^{R(\tau_i)}}{Z(\theta)} \tag{3}$$

where the partition function $Z(\theta)$ normalizes the reward function over all possible trajectories. After taking logarithm of both sides of (3), we get

$$\log P(\tau_i|\theta) = R(\tau_i) - \log Z(\theta), \tag{4}$$

and noting that the partition term $Z$ is constant:

$$\log P(\tau_i|\theta) \propto R(\tau_i). \tag{5}$$

Due to the linearity from Eq. 2, the cumulative reward of a trajectory can then be defined as

$$R(\tau) = \theta^T \phi(\tau), \tag{6}$$

where $\phi(\tau) = \sum_{\mathbf{s_i} \in \tau} \phi(\mathbf{s_i})$. Thus, the log probability of trajectories is proportional to a linear combination of features that compose the reward.

$$\log P(\tau_i|\theta) \propto \theta^T \phi(\tau_i) \tag{7}$$

Based on this proportionality, to determine whether a specific feature contributes to the reward, we evaluate the correlation between feature's expectation and log probability of trajectories. Since ground truth rewards are not available, we use log probabilities of trajectories as pseudo-labels.

## 4.2 Pseudo labels

We generate pseudo labels by computing log probabilities of trajectories empirically from training data $D = \{\tau_1, \tau_2, \ldots, \tau_k\}$. The probability of trajectory $\tau = \{s_1, s_2, \cdots, s_n\}$ is given as

$$P(\tau) = P(s_1)P(s_2|s_1)P(s_3|s_2)\ldots P(s_n|s_{n-1}). \tag{8}$$

We simplify the computation of conditional probabilities by decomposing them into the marginal and joint probabilities of consecutive states. After applying the logarithm to both sides and simplifying the terms, we get

$$\log P(\tau) = \log P(s_1) + \sum_{t=1}^{n-1} (\log P(s_t, s_{t+1}) - \log P(s_t)) \tag{9}$$

The derivation of the Eq. 14 is included in the Appendix D.1. Using the entire training dataset, we perform multivariate kernel density estimation (KDE), to obtain marginal $P(s)$ and the joint density of sequential states $P(s_t, s_{t+1})$. KDE provides a non-parametric way to estimate the probability density function of the states and requires substantial less training data, unlike neural networks.

## 4.3 Candidate features

To perform feature selection using Eq.7, a finite set of candidate state features is required, as evaluating all possible features may be computationally infeasible. According to the moment matching method [41], the probability of the state $\mathbf{s}$ can be approximated by their higher moments. Using moments up to the third order, we can approximate state probability as

$$\log P(\mathbf{s}) \approx \Phi_1(\mathbf{s}) + \Phi_2(\mathbf{s}) + \Phi_3(\mathbf{s})$$

where $\Phi_1(\mathbf{s})$ represents the first-order moments (mean), $\Phi_2(\mathbf{s})$ represents the second-order moments (covariance), $\Phi_3(\mathbf{s})$ represents the third-order moments (skewness), $\mathbf{s}$ is a state vector. As proposed features are evaluated on their predictive power of the log probabilities of trajectories, which in turn consist of state trajectories, we include all covariance terms in the candidate set of features $\Phi(\mathbf{s}) \in \{\Phi_1(\mathbf{s}), \Phi_2(\mathbf{s}), \Phi_3(\mathbf{s})\}$. However, due to the curse of dimensionality and the finite set of demonstrations, using all features as reward components is impractical. Feature matching with IRL may fail due to noise and spurious correlations, leading to false correlations that do not exist in the original data distribution. Therefore, we present an efficient method to select only the relevant features.

## 4.4 Feature selection

After computing the probability and feature expectations of trajectories, we obtain the following data.

$$X = \begin{bmatrix} \phi_1(\tau_0) & \phi_2(\tau_0) & \cdots & \phi_K(\tau_0) \\ \phi_1(\tau_1) & \phi_2(\tau_1) & \cdots & \phi_K(\tau_1) \\ \vdots & \vdots & \ddots & \vdots \\ \phi_1(\tau_N) & \phi_2(\tau_N) & \cdots & \phi_K(\tau_N) \end{bmatrix} \quad Y = \begin{bmatrix} \log P(\tau_0) \\ \log P(\tau_1) \\ \vdots \\ \log P(\tau_N) \end{bmatrix}$$

One way to perform feature selection is to utilize a univariate feature selection method based on statistical tests. First, for each feature separately, we apply F-statistics between trajectory

feature expectations and trajectory probabilities, i.e., between each column of $X$ and the vector $Y$. This gives the feature importance metric, which we can use for ranking the features. Then, the features with higher F-statistics are selected and included in the feature extractor $\phi(\cdot)$. Univariate feature selection is more suitable when the number of samples is much smaller ($K \gg N$) than the size of the candidate set, which is often the case for high-dimensional states. Therefore, directly using linear regression for $X$ and $Y$ results in overfitting. Recursive feature elimination methods may be used here, but they are computationally demanding.

Although the univariate feature selection method does not account for interactions between features, these interactions will be considered while learning the reward function through inverse reinforcement learning. Additionally, to reduce overfitting, we employ cross-validation and noise addition methods. Specifically, we add multiple non-expert trajectories by labeling them with uniformly sampled log probability values from the bottom 10th percentile of expert log probabilities. The time complexity of the feature selection algorithm is $\mathcal{O}(N)$ with respect to the size of the candidate set $\Phi$. This approach is efficient, as it does not involve iterative search of features and policy learning, which would be the case with manual feature construction. The next step to fully recover the reward function is to find the weights of each of the features, which is covered in the next part.

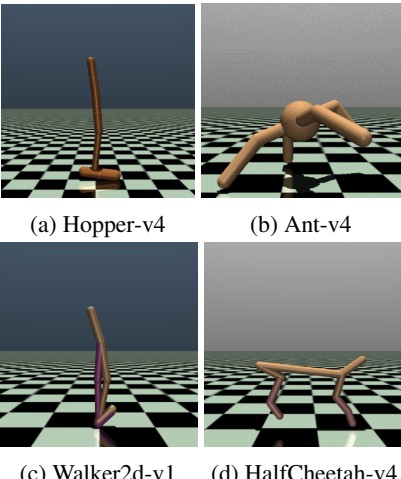

(a) Hopper-v4  (b) Ant-v4

(c) Walker2d-v1  (d) HalfCheetah-v4

Figure 2: Benchmark tasks used in this paper.

### 4.5 Reward and policy retrieval

We employ the maximum entropy IRL [19] to learn the weights of the selected features. To maximize the log probability of the observed data, we formulate the optimization problem as follows:

$$\theta^* = \arg\max_{\theta} \sum_{\tau \in D} \log P(\tau|\theta) \tag{10}$$

To find the optimal weights $\theta$, we take the derivative of log-likelihood that is given by:

$$\nabla L(\theta) = \nabla \sum_{\tau \in D} \log \left( \frac{e^{\theta^T \phi(\tau)}}{Z(\theta)} \right) = \sum_{\tau \in D} \phi(\tau) - \sum_{\tau \in \mathcal{T}} p(\tau|\theta)\phi(\tau) \approx \mu_e - \sum_{i=1}^{m} \sum_{\mathbf{s_i} \in \tau} \phi(\mathbf{s_i}) \tag{11}$$

Here, $\mu_e$ represents the feature expectation of the training data, $m$ is the number of trajectories from current policy. The weights of the reward can then be updated by using the gradient descent as follows:

$$\theta \leftarrow \theta + \alpha \nabla L(\theta) \tag{12}$$

where $\alpha$ is the learning rate. The full derivations are included in Appendix D.2. The learned reward function has an explicit form allowing us to apply any value iteration algorithm to obtain a policy. We utilize the Soft Actor-Critic (SAC)[42] to learn the expert policy by optimizing Eq. 1.

## 5 Experiments

In this section, we consider different continuous control tasks and learn their reward functions from expert data. Then, using the learned rewards, we learn the policy and compare it against the expert performance. We will investigate the following research questions:

1. Is the proposed methodology effective in reconstructing ground truth rewards solving various robotic tasks?
2. Are the learned rewards transparent and amendable?

|  | Linear | Random | Manual | **Proposed** | Expert |
|---|---|---|---|---|---|
| Ant | 1681.47±1000.39 | -777.41±372.42 | 1095.32±552.3 | **2236±987.46** | 1970.048 ± 938 |
| Hopper | 1222.38±606.11 | 470.65±815.62 | 974.89±382.07 | **2110±197.57** | 2901.09±16.09 |
| Walker2d | 1157.47±382.79 | 45.50±150.99 | 1233.16±208.76 | **3108±296.14** | 4241.83±31.07 |
| HalfCheetah | 4922.94±903.50 | -213.11±598.35 | 4991.05±149.02 | **5514±204.02** | 6806.2±113.35 |

Table 2: Mean cumulative rewards for policies trained using various feature sets, calculated across 10 test simulations under varying initial conditions. The last column shows the expert RL policy with *ground truth* reward of the environment. The cells with bold values indicate highest scores.

## 5.1 Setup

We conduct experiments on different environments with continuous state space, particularly Mu-JoCo environments such as Ant, HalfCheetah, Walker2d, and Hopper from the Gymnasium [43] benchmark (see Fig.2). The descriptions of the tasks and ground truth reward formulations are provided in Appendix A.

## 5.2 Data collection

To collect the expert or training data, we train RL algorithms for the above environments. Specifically, we used SAC for all the environments with a continuous action space. We train RL algorithms to maximize the cumulative *ground truth* reward function of the environment. After reaching established benchmark results as presented in Stable Baselines3 [44] for these tasks, we refer to the RL policy as an expert policy and deploy it for data collection. For each of the environments, we collected $N$ trajectories with random starting states, and the simulated trajectories were saved in dataset $D$. Thus, in our IRL method, as shown in Algorithm 1, only dataset $D$ is used, and we assume that the expert policy or ground truth reward function is unknown. We trained the SAC algorithm for a very limited time–40,000 simulation steps–to generate sensible but suboptimal(non-expert) trajectories for feature selection phase. More implementation details are provided in Appendix B.

## 5.3 Baselines

In this work, we propose a feature selection mechanism aimed at recovering the reward function that corresponds to the expert's behavior, given the dataset. Therefore, we compare our method against various baseline feature selection strategies: *manual* features, *randomly* selected features, direct use of all states as features (referred to as *linear* features). For manual feature construction, we evaluated several hand-crafted features and selected the best ones for final experiment. Similarly, the random projection method evaluates a random subset of features from the candidate set. Given that comparing reward functions directly across baselines does not yield a meaningful performance metric in terms of completing a task, we focus on comparing the *policies derived* from these reward functions. To this end, we conduct two comparative analyses. First, we execute the derived policies in multiple testing environments configured by *ground truth* reward functions and observe cumulative rewards. This evaluation is valid because our training data comes from an expert trained with *ground truth* reward function as well. Second, we compare the state distributions of the training data from expert and testing data from the extracted policies. For the divergence measure of multivariate distributions, we exploited the 2D Wasserstein distance metric [45]. Also, we compared the performance of the method against SOTA adversarial inverse reinforcement learning methods such as AIRL[27] and GAIL [35].

## 6  Results

The cumulative episodic rewards of policies trained using different rewards using baseline methods are shown in Table 2. The results show the average for 10 trajectories with three random seeds. The per-

| Task | HalfCheetah | Walker2d | Hopper | Ant |
|---|---|---|---|---|
| Advers. | 609 | 609 | 609 | 609 |
| Ours | 12 | 16 | 10 | 20 |

Table 1: Number of reward network parameters.

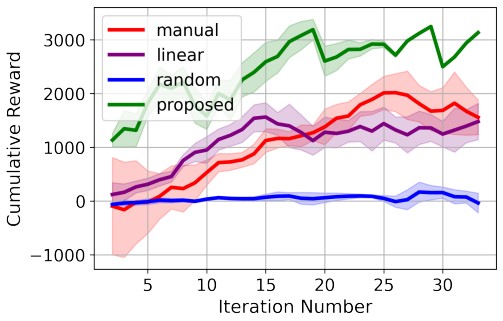
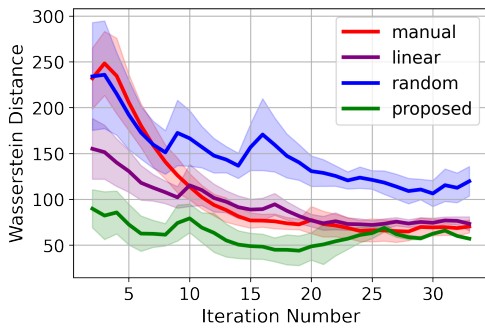

(a) Mean cumulative rewards of the learner using different baseline methods.

(b) 2D Wasserstein distance between state distributions of expert and learner.

Figure 3: IRL training curves for the Walker2d task comparing different feature selection baselines.

formance of the proposed reward model outperforms the baselines for all of the tasks.

Next, we compare the performance of our method to AIRL and GAIL algorithms from the imitation learning library [46]. Table 3 presents the mean cumulative rewards for the tasks studied. We can see that our method achieves higher cumulative rewards in all domains, with significant improvement on the Walker2d, Hopper, and Ant tasks. While adversarial IRL methods achieve state-of-the-art (SOTA) performance, they require a large amount of training data to perform well. Table 1 shows the number of parameters in the reward model. In adversarial IRL methods, a two-layer MLP with 32 nodes is used, while proposed method uses linear reward model with significantly less parameters. This enables to simplify learning process of the reward and therefore, proposed method performs better than adversarial methods with less data.

| Task | HalfCheetah | Walker2d | Hopper | Ant |
|------|-------------|----------|--------|-----|
| GAIL | $2186 \pm 287$ | $872 \pm 336$ | $628 \pm 59$ | $-389 \pm 229$ |
| AIRL | $5264 \pm 88$ | $1856 \pm 89$ | $1289 \pm 165$ | $-1189 \pm 570$ |
| Ours | $5514 \pm 204.02$ | $3108 \pm 296.14$ | $2110 \pm 197.57$ | $2236 \pm 987.46$ |

Table 3: Mean cumulative rewards for different IRL algorithms

The learning curves of the IRL for Walker2d are shown for different reward models in Fig. 3a. Similarly, the 2D Wasserstein distance metric [45] between expert and learner state distributions during training is shown in Fig. 3b. To compute the divergence metric, we use 40 trajectories of expert and learner. The divergence metric between expert and learner policies clearly demonstrates that for the proposed reward, not only does the policy achieve high cumulative rewards, but it also bet-

| | Expert | Fast | Slow |
|---|--------|------|------|
| $v_x$ | $0.12 \pm 0.1$ | $0.17 \pm 0.18$ | $0.09 \pm 0.08$ |

Table 4: Velocity of the HalfCheetah along the x-axis changes accordingly after modifying the weights of the reward component $s_8$.

ter matches the expert state distribution compared to other baseline methods. Next, Fig. 4a shows the correlation plot between ground truth rewards and recovered rewards for 150 trajectories with random initial conditions. The Pearson correlation score is reported in the figure for the HalfCheetah environment. We observe a strong correlation between recovered and ground truth rewards, with a Pearson correlation coefficient of 0.78. This finding is noteworthy, as no ground truth reward signal was provided during reward learning; only expert demonstrations, specifically state trajectories, were used to infer rewards. This suggests that the obtained rewards not only result in behaviors similar to those of the expert but also achieve this by constructing a reward signal that closely resembles the ground truth reward signal. In addition, Fig. 4b shows the full RL training loop with the recovered reward, suggesting the compatibility of the recovered rewards with the RL framework, meaning that RL can be trained from scratch using this reward function.

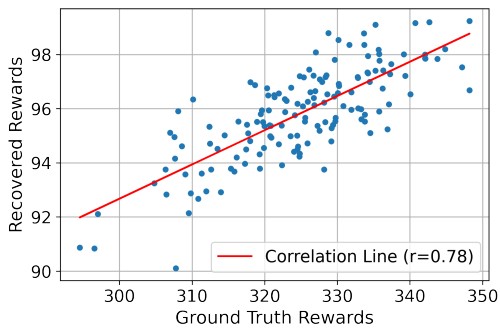

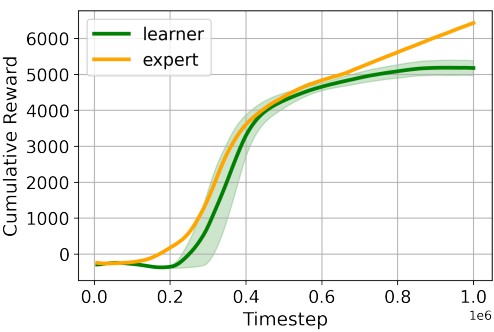

(a) Ground truth and recovered cumulative episodic rewards for trajectories in the expert data

(b) RL learning curve with ground truth reward (expert) and recovered reward functions (learner).

Figure 4: Comparison of training with ground truth and recovered reward functions for HalfCheetah.

Next, we examine the characteristics of the rewards obtained. Let's consider the inferred reward model for HalfCheetah:

$$R = (1.59 - 0.03s_0 + 0.57s_{11} - 0.06s_{16} + 0.21s_0^2 + 0.21s_0s_{11})s_8$$
$$+ (0.31s_1 - 0.38s_2 - 0.83s_0s_2)s_6 - 1.24s_9^2 + 1.03s_{10}^2s_5 + 0.57s_1s_3$$

The transparency of the reward can be seen from the reward representation which consists of monomials. This enables the analysis of which features of the expert contributes to reward and interaction between state features. We observe that the reward terms focus on state $s_8$, which represents the velocity along the $x$-axis, directly relating to the agent's original reward. Additionally, there is a high negative gain for $s_9$, the angular velocity along the $y$-axis, which is sensible since the HalfCheetah should primarily move along the $x$-axis. Negative gains for the angular velocity of the rotors, specifically $s_2$ and $s_{16}$, are also evident, aligning with the control penalty in the ground truth reward.

One important benefit of this compact and explicit representation of the reward is the possibility of fine-tuning the reward function to generate different behaviors. The Table 4 below shows the mean and standard deviation of the HalfCheetah's velocity along the x-axis after multiplying coefficient of one of the terms, $s_8$, in the reward function by 2 and 0.5, respectively. As a result of amending reward components, faster and slower motion was achieved, which can be expected as state $s_8$ accounts for the velocity along the x-coordinate of the front tip.

One limitation of this paper is its applicability to other high-dimensional inputs, such as images. We anticipate that pre-learning features, possibly through the use of autoencoders, and combining with proposed method could be a viable approach to overcoming this limitation, and we plan to explore this direction in future work. Another limitation is our experimental design. We have not conducted experiments with real robots and human demonstrations. However, we validated our method using complex robotic environments. In future work, we plan to apply the method to real-world tasks and learn rewards from demonstrations.

## 7 Conclusion

This paper presented a method for constructing a transparent reward model with an explicitly represented form using non-linear state features. Reward components were selected using an unsupervised learning method from observational data by incorporating the maximum entropy formulation for the behavior. As shown above, selecting the features that allow predicting trajectory probabilities results in a higher match with ground truth rewards and enables the retrieval of policies more similar to those of an expert. Learning rewards from data alleviates the challenge of manual reward specification for complex tasks and allows for training with exploration outside of the training data using the reinforcement learning framework. We have demonstrated that across different environments, recovered rewards outperform other baseline methods.

**Acknowledgments**

This work was supported by the Academy of Finland under grant 347199. The authors would like to acknowledge the computational resources provided by CSC - IT Center for Science and Aalto Science-IT project.

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

# Appendix

## A  Details of Experiments

Below are the short descriptions of the benchmark tasks and the corresponding *ground-truth* reward functions:

1. **Hopper-v4** The task of this one-legged robot is to move forward by applying torques to three hinges. The **true** reward is calculated using distance moved forward and high torque values are penalised.

2. **Ant-v4** The task of this robot consisting of three links is to move forward by applying torques to rotors. The **true** reward is calculated using distance moved forward and high torque values are penalised.

3. **Walker2d-v4** The task of this robot is to move forward by applying torques to its six hinges. The **true** reward is calculated using distance moved forward and high torque values are penalised.

4. **Half-Cheetah-v4** The task of this two-dimensional robot consisting of 9 body parts is to move forward by applying torques to joints, hinges, and feet. The **true** reward is calculated using distance moved forward and high torque values are penalised.

| Tasks | $dim(\mathcal{S})$ | $dim(\mathcal{A})$ |
|------------|------|------|
| Hopper | 11 | 3 |
| Walker | 17 | 6 |
| HalfCheetah | 17 | 6 |
| Ant | 27 | 8 |

Table 5: State-action dimensionality of the tasks

An additional challenge in reward learning is that the features composing the *ground-truth* rewards are not directly available. For example, the direct components of the ground-truth reward, particularly the $x$-coordinate and torque values, are hidden from the states and, consequently, from the reward model. Instead, we infer the reward from the available indirect features like velocity or joint angles.

## B  Implementation details

The source code was implemented in Python 3.8, and the source code will be publicly available upon acceptance. We utilized the Stable-baselines3 library [44] for the training of RL algorithms. As a policy network, we used a multi-layer perceptron (MLP) with two hidden layers. Furthermore, we conducted a thorough hyperparameter optimization for both the RL and inverse reinforcement learning (IRL) parameters. A table with the hyperparameters used during the training is included in the supplementary materials. The total number of training iterations for IRL is 50, and the training dataset consists of 150 trajectories. To facilitate the training, we parallelized data collection and environment simulation using multi-processing techniques. During the feature selection process, we normalize each state feature to have a mean of zero and a standard deviation of one. We apply the same normalization for reward computation. However, for policy learning, we use the raw state inputs. We employed the Optuna[47] library with grid search to tune hyperparameters, and Adam optimizer[48] was used to learn reward model.

## C   Algorithm

---

**Algorithm 1:** Inverse Reinforcement Learning with Unsupervised Feature Selection

---

**input** : $\mathcal{M} \backslash r$, expert data $D$, simulator $E$, iterations $M$, learning rate $\alpha$, number of trajs $N$
**output:** Reward $r$, policy $\pi$

1  $\Phi \leftarrow$ Generate candidate feature set
2  Generate pseudo labels $Y$
3  Compute feature expectations $X$
4  $H \leftarrow$ rank_features$(X, Y)$
5  $\phi(\cdot) \leftarrow$ select_topk$(H, k)$
6  $\mu_e \leftarrow$ compute_features$(D, \phi)$
7  Initialize $\theta \sim Unif[-1, 1]$
8  **for** $i = 0$ **to** $M$ **do**
9      $r \leftarrow$ construct_reward$(\theta, \phi)$
10     Configure simulator $E$ with new $r$
11     $\pi \leftarrow$ learn_policy$(E, r)$
12     $G \leftarrow$ collect_rollouts$(\pi, E, N)$
13     $\mu_a \leftarrow$ compute_features$(G)$
14     $\nabla L(\theta) = \mu_e - \mu_a$ ;                                          // loss gradient
15     $\theta \leftarrow \theta + \alpha \nabla L(\theta)$ ;                          // update $\theta$
16  **end**

---

## D   Proofs

### D.1   Computation of trajectory probability

The probability of trajectory $\tau = \{s_1, s_2, \cdots, s_n\}$ is given as

$$P(\tau) = P(s_1)P(s_2|s_1)P(s_3|s_2)\ldots P(s_n|s_{n-1}). \tag{13}$$

After applying the logarithm to both sides and simplifying the terms, we get

$$\log P(\tau) = \log P(s_1) + \sum_{t=1}^{n-1} \left( \log P(s_t, s_{t+1}) - \log P(s_t) \right) \tag{14}$$

Here, conditional state probability $P(s_{t+1}|s_t)$ can be written as $P(s_{t+1}|s_t) = P(s_t, s_{t+1})/P(s_t)$. Then:

$$P(\tau) = P(s_1)\frac{P(s_1, s_2)}{P(s_1)}\frac{P(s_2, s_3)}{P(s_2)}\cdots\frac{P(s_{n-1}, s_n)}{P(s_{n-1})} \tag{15}$$

After applying the logarithm to both sides, we get:

$$\log P(\tau) = \log P(s_1) + \log \frac{P(s_1, s_2)}{P(s_1)} + \log \frac{P(s_2, s_3)}{P(s_2)} + \ldots + \log \frac{P(s_{n-1}, s_n)}{P(s_{n-1})} \tag{16}$$

$$= \log P(s_1) + \sum_{t=1}^{n-1} \log \frac{P(s_t, s_{t+1})}{P(s_t)} \tag{17}$$

$$= \log P(s_1) + \sum_{t=1}^{n-1} \left( \log P(s_t, s_{t+1}) - \log P(s_t) \right) \tag{18}$$

$$\tag{19}$$

### D.2   Learning of reward weights

$$\theta^* = \arg\max_{\theta} \sum_{\tau \in D} \log P(\tau|\theta) \tag{20}$$

To find the optimal weights $\theta$, we take the derivative of log-likelihood that is given by:

$$\nabla L(\theta) = \nabla \sum_{\tau \in D} \log \left( \frac{e^{\theta^T \phi(\tau)}}{Z(\theta)} \right) \tag{21}$$

$$= \nabla \sum_{\tau \in D} \left( \theta^T \phi(\tau) - \log Z(\theta) \right) \tag{22}$$

$$= \sum_{\tau \in D} \phi(\tau) - \nabla \log Z(\theta) \tag{23}$$

$$= \sum_{\tau \in D} \phi(\tau) - \frac{1}{Z(\theta)} \sum_{\tau' \in \mathcal{T}} \phi(\tau') e^{\theta^T \phi(\tau')} \tag{24}$$

$$= \sum_{\tau \in D} \phi(\tau) - \sum_{\tau \in \mathcal{T}} p(\tau|\theta)\phi(\tau) \tag{25}$$

$$\approx \mu_e - \sum_{i=1}^{m} \sum_{\mathbf{s_i} \in \tau} \phi(\mathbf{s_i}) \tag{26}$$

## E  Hyperparameters

|  | HalfCheetah | Walker | Hopper | Ant |
|---|---|---|---|---|
| Number of expert trajs | 150 | 150 | 100 | 150 |
| Number of iterations | 50 | 50 | 50 | 50 |
| Learning rate | 0.05 | 0.03 | 0.05 | 0.03 |
| Learning decay | 0.985 | 0.99 | 0.99 | 0.985 |
| RL timestep | 1e6 | 1e6 | 1e6 | 1e6 |
| Discount factor | 0.99 | 0.99 | 0.99 | 0.99 |
| Number of parallel envs | 4 | 4 | 4 | 4 |
| Batch size | 1024 | 256 | 512 | 256 |
| Number of nodes (MLP) | 256x2 | 256x2 | 256x2 | 256x2 |

Table 6: Training hyperparameters.

