# OpenReview forum: "Learning Transparent Reward Models via Unsupervised Feature Selection"
_robot-learning.org/CoRL/2024/Conference — CoRL 2024_

### Official Review · Reviewer_RAMo · 2024-07-14
**Not ready for publication**

**Originality:** 2
**Technical Quality:** 2
**Clarity Of Presentation:** 2
**Potential Impact:** 2
**Recommendation:** 3
**Confidence:** 3

**Review:**

## Strengths

**Method:** The method seems principled and incorporates traditional machine learning techniques for deep IRL

**Results:** Results are not bad, demonstrating the method is better than ablations and (somewhat unconvincingly with no standard deviations) that the method possibly outperforms AIRL.

## Weaknesses

**Writing:** I have quite a few complaints about some of the writing and clarity of the paper:

- The first paragraph of the intro doesn’t hint at or make clear what problem the paper is trying to solve.
- Combined with the above point, there are no method or teaser figures that clearly demonstrate the problem setting (inputs/outputs) and method at a high level. This makes it hard to understand what the paper will be about until the end of the intro.
- Interpretability isn’t talked about much throughout the paper yet it’s highlighted in the title and briefly mentioned in the intro. More justification should be given for how this method is more interpretable than other methods that learn a linear function of the state features as a reward.
- Notation: the notation for how to write conditional probabilities seems highly nonstandard: $P(\tau_i)|\theta)$ is written many times but to me it just looks like there’s an extra parenthesis after $\tau_i$.
- Section 4.2: why kernel density estimation in particiular? no intuition is included of using this versus any other method of estimating probabilities (e.g., train a neural network)
- Section 4.3: Why the 3rd order specifically? the experiments don’t justify this choice over 4th order or just 2nd order?

**Experiments:**

- How many seeds were used? This should be specified
- The AIRL baseline is from 2017, even though it’s a valid baseline. Perhaps a few more recent baselines would make sense to compare to.
- There's no standard deviations for the AIRL baseline either, so these results are unconvincing. Furthermore, it's only presented on half the tasks.
- Figure 2: what’s an epoch on the x axis? I’d recommend remaking these WandB plots using their API to download the stats and plotting them with proper labels and bigger font sizes.
- There’s some missing analysis in the experiment results, e.g. L 241: why is the proposed method better than AIRL here?
- How are hyperparameters tuned for the proposed method and ablations/baselines?
- Not much is done with respect to the interpretability claim. One could perform experiments demonstrating that the form of interpretable reward functions is actually amenable to editing/preferred by humans/etc, e.g., Section 5.7 of this [paper](https://clvrai.github.io/leaps/).

Limitations: The authors should include some discussion of the limitations in the paper.




POST REBUTTAL:

A lot of my concerns have been addressed, and after looking at the revised paper, I am updating my score to a weak accept. I think the paper has merit.

**Quality Of The Limitations Section:**

1

**Questions For Rebuttal:**

See above questions in weaknesses.

**Robotics Focus:**

3

**Summary Of Paper:**

This paper proposes a feature selection method for IRL which extracts state features and values highly correlated with the expert behavior through an explicit feature extraction and density estimation step.

**Summary Of Recommendation:**

Not ready due to some substantial writing issues, a lack of new baselines, and insufficient experiments.

---

### Official Review · Reviewer_rzLp · 2024-07-19

**Originality:** 3
**Technical Quality:** 3
**Clarity Of Presentation:** 3
**Potential Impact:** 3
**Recommendation:** 3
**Confidence:** 4

**Review:**

This paper studies the important problem of learning intepretable reward models. The motivation is well done and the idea of using log probabilites of trajectories to find an appropriate set of candidate features is novel as far as I can tell.

Is there a reason to estimate the marginals and joint probability distributions as opposed to estimating the conditional distributions?

Noise addition method is unclear. How do you use non-expert trajs? More details would be good.

The paper states that the method does not include many combinations of features. It's unclear what this means and whether it is a good or bad thing. It seems that you would want to be able to learn non-linear features that are combinations of state features.

Eqns 11-14 seem to be identical to the Max Ent original paper so you could cut space there by not including the derivation.

It would be nice to have more details on the manual feature construction baseline as there are many ways to do this. How were the features determined? What kind of trial and error process was used?

Furthermore, it's unclear what kinds of features your method uses. Are they just raw state dimensions? Is there any preprocessing? How would this work with visual inputs where you wouldn't want to have each pixel be a feature? It seems that the reward is just a function of raw state, but the reward shown on line 228 is a non-linear function of state and it's unclear to me how this type of reward is obtained. Have the authors considered feature learning approaches that return a linear reward function such as
- Brown et al. "Safe imitation learning via fast bayesian reward inference from preferences." ICML, 2020.

Figure 2 is nice in showing the behavioral similarity between the expert and the learned policy when using the proposed approach.

Why only compare against AIRL on two domains? It would be good to include the AIRL comparison in the Table 1 across all domains. Also other baseline approaches such as "Feature construction for inverse reinforcement learning" would be good to compare against if possible.

The idea of interpretable rewards is very appealing, but looking at the half-cheetah reward on line 228 it is hard to tell how interpretable it is (it doesn't seem very interpretable to my untrained eye). It would be nice to actually study the interpretability and to compare with other interpretability approaches for reward learning.

Overall the results in Table 1 and 2 show that the proposed approach works better than other approaches, but some of the baselines are unclear such as the manual and random projection. Comparing with AIRL and maybe Guided cost learning would be good as well as comparing against other feature selection/interpretable reward learning approaches.





Several typos and formatting issues throughout.
- log P(\tau_i) | \theta) ->  log P(\tau_i | \theta)
- line 112 "states ss"
- Reference [27] has the wrong authors for the AIRL paper.
- line 236 has a spurious newline

**Quality Of The Limitations Section:**

1

**Questions For Rebuttal:**

See questions above.

**Robotics Focus:**

3

**Summary Of Paper:**

This paper seeks to learn interpretable reward functions from demonstrations

**Summary Of Recommendation:**

The paper studies an important area and has some promising results but there are insufficient details to fully evaluate and the claim of interpretability requires more validation.

---

### Official Review · Reviewer_psPt · 2024-07-20
**Interesting research on feature selection for reward learning but experimental results are not very impressive.**

**Originality:** 3
**Technical Quality:** 3
**Clarity Of Presentation:** 2
**Potential Impact:** 3
**Recommendation:** 3
**Confidence:** 4

**Review:**

**Strengths**

- S1: Interesting research direction. The idea of automatically selecting state features to develop effective and interpretable reward models is compelling. To achieve this, the authors proposed a new solution based on trajectory density estimation from expert demonstrations.

- S2: Well-designed experiments. The experiments are thoughtfully designed to demonstrate the benefits of the proposed method. In addition to policy learning results, the authors provide insightful analysis on interpretability and state distribution matching.

**Weaknesses**

- W1: The authors used simple simulated robots like Hopper, Ant, Walker, and HalfCheetah in their experiments. This limits the impact of the experimental results. It would be beneficial if the authors could show results using robots with more joints (such as Humanoid), which would involve a higher-dimensional state space.

- W2: Several components are hard to understand: (1) The explanation in lines 98-100 is a bit unclear. This section is crucial for explaining the motivation and intuition behind using log probabilities as pseudo-labels. It would be helpful if the authors could elaborate more on this point. (2) The process of creating candidate features is unclear. The authors introduced moment matching, but it is not clear what $\phi$ represents exactly. More details on this aspect would be beneficial.

**Quality Of The Limitations Section:**

3

**Questions For Rebuttal:**

**Random seeds**

- The authors mentioned, "The results show the average for 10 trajectories with random seeds." How many random seeds were used?

**Writing**

- line 112: change "states ss" to "state $s$"

- Eq3,4,5,7: change $P(\tau_i)|\theta)$ to $P(\tau_i|\theta)$

- Algorithm 1, line6: change $(D,\phi$ to $(D,\phi)$

- Eq 1: $t$ notation is missed inside of expectation.

**Robotics Focus:**

3

**Summary Of Paper:**

For learning a reward from human demonstrations, the authors propose a new method to select important state features. Based on max entropy formulation, the authors use trajectory density as labels and find best features to model the trajectory density.

**Summary Of Recommendation:**

The paper explores an interesting direction, but some details are not clear. Additionally, the experimental results are somewhat limited. Therefore, I recommend a 'weak reject.'

---

### Author Rebuttal · Authors · 2024-08-13

We would like to thank the reviewers and area chair for their insightful and thorough comments. We are glad that the reviewers found our work compelling and novel. Following the reviewers' suggestions, we've made the following important changes to improve the quality of the paper:

- We changed the term "interpretable" in the paper to "transparent", which we believe better describes our current results. We agree with the reviewers that interpretability can have different context and, if claimed, requires more evaluation with suitable interpretability metrics. Our proposed method learns a compact, explicit, and amendable reward model from observational data, which we refer to as a transparent reward model in the revised paper.
- **Additional baseline** method was added to compare against the proposed one.
- We added more details of the method, baselines, and implementation.
- We updated the **introduction** to clarify the problem setting and added a figure to illustrate the high-level concept of the method.
- We added the **limitations** to Section 6.
- We revised the figures to improve quality and fixed the typos.

We uploaded the revised manuscript and highlighted the changes or new parts with **blue** color.

---

### Decision · Program_Chairs · 2024-09-04

**Decision:**

Accept

**Comment:**

**Summary**

The paper introduces a method for learning transparent reward functions from human demonstrations by focusing on selecting important state features. Using a maximum entropy framework, the approach involves estimating trajectory density and identifying the features that best model this density. This feature selection method for IRL aims to extract state features that are highly correlated with expert behavior, enhancing the transparency of the learned reward functions.

**Summary of Reviews**

*Strengths*:
- Reviewers found the idea of automatically selecting state features for reward models compelling, novel, and principled.
- The experiments were well-designed to demonstrate the benefits of the proposed approach.

*Weaknesses*:
- Lack of clarity. All reviewers criticized the paper for its lack of clarity, missing crucial details necessary for understanding the method, the baselines, and experimental details.
- Missing technical and experimental details. All reviewers had lingering questions about the method and experiments. The authors should resolve any remaining confusions in the rebuttal.
- Insufficient experiments. Some reviewers complained about the narrow evaluation on just the Hopper, Ant, Walker and HalfCheetah environments. Others asked for more baseline comparisons to AIRL and GCL. The lack of real-world robotics experiments is concerning.
- Missing discussion of limitations.

**Outcome and Post-rebuttal**

Two reviewers substantially improved their scores and indicated their concerns being mostly if not fully addressed. I thank the authors for such a substantial and compelling rebuttal effort! Reviewers found the manuscript significantly improved, and appreciated the reframing as transparency and not interpretability. I recommend Accept for poster and I suggest the authors follow the remaining reviewer suggestions to improve their manuscript for the camera ready. In particular: more details for reproducibility, more details about the handcrafted features, a comparison with FIRL, and a code release.